# Complex Characterization of Germline Large Genomic Rearrangements of the *BRCA1* and *BRCA2* Genes in High-Risk Breast Cancer Patients—Novel Variants from a Large National Center

**DOI:** 10.3390/ijms21134650

**Published:** 2020-06-30

**Authors:** Anikó Bozsik, Tímea Pócza, János Papp, Tibor Vaszkó, Henriett Butz, Attila Patócs, Edit Oláh

**Affiliations:** 1Department of Molecular Genetics, National Institute of Oncology, Ráth György utca 7-9, H-1122 Budapest, Hungary; timi.pocza@gmail.com (T.P.); hmlh1.hmsh2@gmail.com (J.P.); vaszko.tibor@chello.hu (T.V.); henriettbutz@gmail.com (H.B.); attilapatocsb@gmail.com (A.P.); e.olah@oncol.hu (E.O.); 2Hereditary Cancers Research Group, Nagyvárad tér 4, H-1089 Budapest, Hungary; 3Department of Laboratory Medicine, Semmelweis University, Nagyvárad tér 4, H-1089 Budapest, Hungary

**Keywords:** *BRCA1*, *BRCA2*, large genomic rearrangement, familial breast cancer, copy number analysis, breakpoint characterization, deletion, duplication

## Abstract

Large genomic rearrangements (LGRs) affecting one or more exons of *BRCA1* and *BRCA2* constitute a significant part of the mutation spectrum of these genes. Since 2004, the National Institute of Oncology, Hungary, has been involved in screening for LGRs of breast or ovarian cancer families enrolled for genetic testing. LGRs were detected by multiplex ligation probe amplification method, or next-generation sequencing. Where it was possible, transcript-level characterization of LGRs was performed. Phenotype data were collected and analyzed too. Altogether 28 different types of LGRs in 51 probands were detected. Sixteen LGRs were novel. Forty-nine cases were deletions or duplications in *BRCA1* and two affected *BRCA2*. Rearrangements accounted for 10% of the *BRCA1* mutations. Three exon copy gains, two complex rearrangements, and 23 exon losses were characterized by exact breakpoint determinations. The inferred mechanisms for LGR formation were mainly end-joining repairs utilizing short direct homologies. Comparing phenotype features of the LGR-carriers to that of the non-LGR *BRCA1* mutation carriers, revealed no significant differences. Our study is the largest comprehensive report of LGRs of *BRCA1/2* in familial breast and ovarian cancer patients in the Middle and Eastern European region. Our data add novel insights to genetic interpretation associated to the LGRs.

## 1. Introduction

Germline pathogenic single nucleotide variations (SNVs) and small indels of the *BRCA1* and *BRCA2* genes are well-studied genetic changes in hereditary breast and ovarian cancers [1]. However, large genomic rearrangements (LGRs), including deletions and duplications affecting whole exons also contribute to the mutational landscape of these genes. Surveys from different populations documented that approximately 10% of the overall *BRCA1/2* germline mutations are LGRs, but the exact ratios are strongly population dependent. LGRs in *BRCA1* are responsible for between 0% and 27% of all *BRCA1* disease-causing mutations identified in numerous populations. Such alterations are far less common in the *BRCA2* gene [2,3,4,5,6,7,8,9,10,11,12,13,14,15,16,17]. These chromosomal changes are not readily detectable by conventional sequencing, but require copy-number sensitive methods, such as multiplex ligation-dependent probe amplification (MLPA) [18], quantitative multiplex PCR of short fluorescent fragments (QMPSF) [19], or comparing the relative numbers of the aligned reads yielded by next-generation sequencing (NGS) [20].

The source of LGRs in the *BRCA1/2* genes is mainly recombination between low-copy paralogue sequences, such as Alu motifs residing in introns [21]. *BRCA1*, with its markedly Alu-dense genomic context is especially prone to non-recurrent rearrangements of unique sizes [22]. Although most LGRs are definitively pathogenic, causing frameshifts and premature termination codons, some rearrangements have ambiguous effects, especially in-frame deletions of redundant exons [23] or some duplications, where additional copies of exons might be tolerated [7] or may as well be posited in a different genomic context.

From 2004, the Department of Genetics of National Institute of Oncology, Budapest, Hungary, has been involved in the *BRCA1/2* germline mutation screening of the enrolled high-risk Hungarian breast and ovarian cancer patients. We performed a comprehensive characterization of LGRs detected in our patients. Exact determination of their position, frequency, and pathogenicity were studied and we also made attempts to decipher the underlying molecular mechanisms of each LGR. Transcript-level evaluations of the variants were done in cases, where RNA samples of the probands could be obtained: this approach provided precious information concerning splicing consequences and allelic expressions. We also analyzed phenotype features observed in families harboring LGRs in order to test whether LGRs confer more severe disease than other small-scale mutations of the gene.

Our study is the first comprehensive report of LGRs of *BRCA1* and *BRCA2* genes in Hungarian familial breast and ovarian cancer patients and the largest survey in the Middle and Eastern European region.

## 2. Results

As part of the routine genetic testing in our department we screened for LGRs affecting whole exons of the *BRCA1/2* genes by MLPA as well as by NGS methods. From 2004, we identified altogether 51 LGRs in index patients of unrelated breast or ovarian cancer families (Table 1, Appendix A). An example for *BRCA1* del(ex18–22) detection by NGS is shown in Appendix A. Of the detected LGRs 49 were variants of *BRCA1* and only two affected *BRCA2*. LGRs of *BRCA1* accounted for approximately 10% of the overall *BRCA1* mutations, whereas *BRCA2* LGRs took up ≤0.5% of the total *BRCA2* mutations.

With the application of serial long-range PCRs and nested sequencings of the breakpoint-containing PCR products (the junction fragments) we determined the exact upstream (5’) and downstream (3’) breakpoints of each rearrangement, where it was feasible. Breakpoint determination with the examples of *BRCA1* del(ex24), *BRCA1* del (ex1-20), and *BRCA1* del(ex8) rearrangements are depicted in Figure 1. Junction position sequencing of all novel LGRs are presented in Appendix A. In all but four cases we managed to determine the exact breakpoints of LGRs. In the remaining cases the breakpoints were also successfully restricted to a more limited interval with semi-quantitative tests. With the help of real-time qPCR and QMPSF tests we pointed out the positions of the actual breakpoints with some hundred base pairs certainty (Figure 2A,B).

Breakpoint characterization unveiled that the 51 unrelated cases made up from 26 different rearrangements (24 deletions and two duplications) in the *BRCA1* gene and two different rearrangements (one deletion and one multiplication) in the *BRCA2* gene (Figure 3, Table 1). Of the 28 different rearrangements, 24 were successfully genotyped for the exact upstream and downstream break positions. Seventeen of them (70.8%) had Alu sequences at both 5’ and 3’ breakpoints. Moreover, six of them had both breakpoints in the same Alu family (AluY-AluY, AluSx-AluSx), that provided even larger sequence homeology. In all cases Alus were in the same orientation for both breakpoints (14 times head-to tail, three times tail-to-head relative to the gene transcript orientation). The average size of the complete homology was 19 bp at the junctions. In six cases we experienced no homeology, and limited or no sequence homology at the breakpoint positions. Two different rearrangements contained short reverse complement insertions beside deletions (complex rearrangements).

LGRs spanned all regions of *BRCA1* gene, no regional hot spots were detected, except for *BRCA1* pseudogene region, which is especially prone for rearrangements. The exact positions of breakpoints inside Alus were also spread evenly throughout the whole Alu region; no seed sequence dedicated for the rearrangements was identified. Interestingly, a *BRCA1* del(ex1-2) variant NG_005905.2:g.59989_96300del had junction point especially inside the polyA tail. The size of deletions ranged from 450 base pairs to several tens of kilobases. In total, seven different rearrangements, where deletion or duplication extended upstream the *BRCA1* gene and also affected the promoter region, were identified.

There were some recurrent LGRs with the same genomic breakpoints: *BRCA1* dup(ex13) LRG_292t:c.4186-1787_4357+4122dup was found in eight families, whereas *BRCA1* dup(ex1-2) NG_005905.2:g.90060_97318dup was detected in three index patients of different families. *BRCA1* del(ex17) LRG_292t1:c.4986+726_5074+84del and *BRCA1* del(ex21-22) LRG_292t1:c.5278-492_5407-128delins236 were detected in four and nine families, respectively.

We contrasted personal and family history as well as some clinical variables of LGR-carriers to those of non-LGR pathogenic *BRCA1* variant carriers. For this latter group we used a cohort of 281 well-characterized samples with non-LGR pathogenic and likely pathogenic mutations identified through routine clinical diagnostic testing. The phenotype features of the LGR-carrier probands and their families are detailed in Appendix A. Neither the personal of familial phenotypes nor the clinical variables showed any significant difference between the LGR and the non-LGR *BRCA1* mutation carriers (Appendix A). Six families, where deletion stretched further upstream the *BRCA1* gene affecting also the *BRCA1* and *NBR2* common promoter or even the *NBR2* of *NBR1* genes themselves did not show more severe disease pathology than LGRs restricted to the *BRCA1* open reading frame. However, this lack of genotype–phenotype correlation might be the consequence of the small number of cases carrying such rearrangements.

Haplotyping of three recurrent rearrangements, *BRCA1* del(ex17), *BRCA1* del(ex21-22), and *BRCA1* dup(ex13) was done with the help of polymorphic STR markers inside and surrounding the *BRCA1* gene recommended by Neuhausen et al. (1996) [31], as well as with SNPs inside the *BRCA1* gene, where it was applicable. All three LGR types, where breakpoint chromosomal positions were the same, proved to be of the same origin according to the genotypes of the respective core haplotypes (Appendix A).

Transcript-level examination of the variant allele was possible in eight families of six different LGRs (see RNA column of Table 1). These involved *BRCA1* del(ex21-22) variants with two different breakpoint positions. PCR on the cDNA templates with primers surrounding the deleted exons yielded shorter extra products in each case, reflecting the successful amplification also from the mutant allele (Figure 4A,B). Sequencing these fragments pointed out that only the deletion-affected exons were missing in all tested LGR types, so exon deletion on gDNA level reflected exactly on RNA level in each case, the function of the neighboring canonical exon splicing positions were not affected. In order to evaluate the relative expression quantity of the two alleles, we performed an allelic imbalance test, where heterozygote positions made it feasible. The ratio of the deletion carrier allele was much less (≤50%) than the normal allele in two deletion types: *BRCA1* del(ex5-10) and *BRCA1* del(ex17) (Figure 4A), but equivalent expression of the two alleles was experienced in *BRCA1* del(ex21-22) (Figure 4B) and *BRCA1* del(ex18-22).

## 3. Discussion

In the course of the routine *BRCA1/2* testing of our department from the year of 2004, we detected 28 different, 16 novel large genomic rearrangements in 51 probands of unrelated breast or ovarian families. According to the statistics based on NGS data of our laboratory, *BRCA1* LGRs take up approx. 10% of the overall *BRCA1* mutations in Hungary. These frequencies are comparable with those detected in other Middle European populations [3,4,8], but fall short of the extreme ratios found in the Netherlands (36% of all *BRCA1* mutants) [15], Italy (19% of all *BRCA1* mutants) [16], and UK (20% of all *BRCA1* mutants) [5], where frequently occurring founder mutations constitute the majority of LGR cases.

The bulk of the rearrangements affected the *BRCA1* gene, *BRCA2* was nearly exempt from LGRs. This is in concert with the findings worldwide (*BRCA1* LGR ratios ranges from 0% to 36%, whereas *BRCA2* LGRs constitute 0–6% of all *BRCA* mutations [2,3,4,5,6,7,8,9,10,11,12,13,14,15,16,17]). The difference is at least partly explained by the larger number of Alus and other repetitive sequences as well as the pseudogene counterpart in *BRCA1* [32]. Theoretically, the relatively low LGR frequency in *BRCA2* might be a consequence of a biased *BRCA2* mutation spectrum due to an over-representation of founder point mutations. However, but the three most frequent point mutations (c.9097dup; c.5946del, c.7913_7917del) represent only 26% of all *BRCA2* mutations in Hungary, which is very close the respective ratios of the countries in our region (e.g., Austria: 23%, Czech Republic: 26%, Poland: 26%). Additionally, the three most frequent founder mutations of *BRCA1* (c.5266dup, c.181T>G, and c.68_69del) account for a much higher ratio (68%) of all *BRCA1* pathogenic variants in our population [17,33], but the ratio of LGRs in *BRCA1* is still higher than in *BRCA2*.

Exact breakpoint characterizations yielded 28 different types of rearrangements. Twelve LGRs with the same breakpoints have been already reported (see Reference column in Table 1). The remaining 16 types of rearrangements have not been described so far, they may be characteristic to the Hungarian population.

There were four rearrangement types (two deletions and two duplications) that occurred recurrently. Neither of them was Hungarian founder mutation, all of them were reported with similar frequencies in other populations (see Table 1 Reference column), with the exception of *BRCA1* del(ex17) LRG_292t1:c.4986+726_5074+84del, which was detected in only one case of 1506 families in Germany [27]. Noteworthy, neither of the two *BRCA1* del(ex1-2) rearrangements was identical with the reported ones, which underpinned that psi*BRCA1* serves as hot spot for recombination.

Among the detected genomic rearrangements deletions prevailed over duplications. This might indicate that the underlying molecular mechanisms are mainly intrachromatidal events between repetitive sequences of the same orientation, which are the main source of copy number loss [34]. The exact breakpoint detections enabled us to make suggestions for the mechanisms that are responsible for the generation of the respective rearrangements. The majority of deletions detected were flanked by Alu sequences sharing ≈300 bp sequence similarity. This deletion type is used to be attributed to homologues recombination repair between ectopic sequences, called non-allelic homologues recombination (NAHR) [21], but novel findings argue that HR events require more extensive homology, than the typical 300 bp of Alu sequences [35]. Over the past decade, microhomology-mediated end joining (MMEJ) and single-strand annealing (SSA) mechanisms were suggested for these deletion types, which are special forms of end joining repair rather than real homologues recombination [36,37,38]. These mechanisms benefit the extensive homeology between two Alu sequences but actually apply only a small uninterrupted homology of 5–50 base pairs [37]. We found six LGR types, where no, or very limited sequence homology was found at the junctions: they may be explained by non-homologues end joining (NHEJ) [39,40]. Two LGR types with complex rearrangements found in our study (deletions combined with reverse insertions) may have arisen from the fork stalling and template switching (FoSTeS) mechanism [41], where the polymerase enzyme stops at defined positions, switches strand to the opposite direction and resumes back after some hundreds of base pairs of reverse strand elongation.

The exact nucleotide positions of the upstream and downstream breakpoints are so characteristic, that they provide sufficient evidence for common origin of the LGRs with the same breakpoints. However, to be more precise, we performed haplotyping of three recurrent LGRs: *BRCA1* del(ex17), *BRCA1* del(ex21-22), and *BRCA1* dup(ex13). Our results confirmed the common origin of the respective LGR types possessing the same breakpoint positions. Collaborative haplotyping of *BRCA1* dup(ex13) samples (referred as ins6kbEx13 in publications) collected from geographically diverse populations was formerly reported by The *BRCA1* Exon 13 Duplication Screening Group [42] as a founder mutation with common origin. Exact sizes of three core STRs of these samples were described by other groups [43,44], which coincided with our results, therefore our samples turned to be identical with the reported ones. Similarly, all but one *BRCA1* del(ex21-22) samples shared the same haplotype as reported by Vasickova et al. (2007) [25]. Core haplotype of our four *BRCA1* del(ex17) samples were also identical, but implicitly was not the same as the recurrent German *BRCA1* del(ex17) samples with different intronic breakpoints haplotyped by Engert et al. (2008) [27].

Transcript-level testing of five deletion types yielded the lack of the respective exons at canonical cassette exon borders on cDNA-level in each case. This indicates that the donor and acceptor sites of the flanking exons were unaffected, irrespective of the size and position of the missing neighboring intron stretches, and no alternative splicing anomalies were detected. This provides inference that the missing intron regions did not contain essential regulation signals for splicing in either case tested.

Benefiting exonic heterozygote *BRCA1* markers of the samples we experienced allelic imbalance of two of the tested four deletion types, *BRCA1* del(ex5-10) and *BRCA1* del(ex17) at cDNA-level. This can be attributed to nonsense-mediated mRNA decay (NMD), since *BRCA1* mutations resulting in premature termination codon (PTC) preceding exon-exon junctions normally trigger NMD and degradation [45]. Indeed, both deletion types harbored PTC. The two other tested deletion types did not show any sign of allelic imbalance. They comply with the NMD-rule, considering *BRCA1* del(ex21-22) is an in-frame deletion without PTC, and *BRCA1* del(ex18-22) has its termination codon in the last coding exon of *BRCA1*, so these samples evade NMD. Although RNA expression data arise from peripheral blood rather than tumor tissue samples of the carriers, data are relevant, since surveys give evidence that *BRCA*-expression in peripheral blood cells is significantly correlated to *BRCA*-mutation carrier status [46].

All LGRs with deletions were ascertained to be clinically significant. In contrast, the pathogenicity of *BRCA1* dup(ex1-2) is still under debate, since it contains an uninterrupted copy of the gene together with its promoter [7,13]. Similarly, pathogenic nature of the *BRCA2* amp(ex21) is also doubtful, because we failed to characterize breakpoint for tandem sequential amplification.

Since LGRs often span several exons and in some cases also affect neighboring functional genes, the question emerged if they are associated with more severe pathology. Considering that the great majority of the LGRs affect the *BRCA1* gene it made sense to compare their phenotype data to those of the non-LGR *BRCA1* mutation carriers. Comparison of several pathological features did not yield a significant difference in any case. Our results confirm that LGRs do not elicit more severe disease than other, small-scale exonic *BRCA1* mutations, in line with other studies [6,28,32,47]. Even the families, where rearrangements affected also the neighboring *NBR2* and *NBR1* genes did not show differences in their phenotype features compared to other LGRs’. Nevertheless, it would be interesting to examine in more cases, if these mutation carriers show any multilocus phenotype, since recent findings argue that *NBR2* is also involved in cancer pathways [48]. There are examples where impairment of disease-associated neighboring genes also influenced the disease severity [49], so we cannot exclude that differences might exist when testing on a larger number of samples.

The survey confirms the necessity for genotyping of LGRs in the course of routine genetic testing, but with caution for the interpretation of the pathology of these types of rearrangements.

## 4. Materials and Methods

### 4.1. Patient Selection and Genotyping

Breast and ovarian cancer patients were enrolled for germline *BRCA1/2* genotyping as part of the routine genetic counseling in the Department of Molecular Genetics of the National Institute of Oncology, Budapest, Hungary. Eligibility criteria were positive family history for either breast or ovarian cancer along with personal clinicopathological features according to the relevant National Comprehensive Cancer Network (NCCN) guideline versions. Research projects and study protocols were approved by the Institutional Ethical Board and the Research and Ethics Committee of the Hungarian Health Science Council (ETT-TUKEB 53720-7/2019/EÜIG, 20 July 2019). All participants provided written informed consent for the genetic testing.

Genomic DNA was isolated from peripheral white blood cells using the Gentra Puregene Kit (Qiagen, Hilden, Germany). The probands were either sequenced by conventional Sanger method on ABI 3130 Genetic Analyzer (Thermo Fisher Scientific, MA, USA) or genotyped by NGS. Sanger sequencing restricted to the exons and exon-intron boundaries of *BRCA1* and *BRCA2* genes. Genotyping by NGS was carried out on Illumina MiSeq platform (Illumina, CA, USA) after an amplicon-based enrichment library preparation of the *BRCA1/2* exons by Multiplicom BRCA MASTR Dx or BRCA MASTR Plus Dx kit (Agilent Technologies, CA, USA). Library preparation was done according to the manufacturer’s instructions and pooled libraries from 24 or 48 indexed patients were run on V2-500 sequencing cartridge system (Illumina). Data analysis form FASTQ files to variant calling was done with Multiplicom MASTR Reporter software (Agilent Technologies) as well as using our in-house bioinformatics workflow.

### 4.2. Gene Dosage Analysis

Detection of whole exon deletions or duplications were performed by MLPA method (MRC-Holland, the Netherlands) with the following probe sets: P002 and P239 for *BRCA1*, P045 for *BRCA2*. To exclude false positive MLPA signals, deletions affecting single exons were reinforced by confirmation probe set (P087 for *BRCA1* and P077 for *BRCA2*) or sequenced to search for possible heterozygote variants inside the probe hybridization positions. Samples genotyped by high-coverage NGS (average read depth exceeding 1000 reads per position) were analyzed for copy number variations by the CNV analysis algorithm of the Multiplicom MASTR Reporter software (Agilent Technologies) based on the relative normalized ratio of the aligned read numbers of each amplicon. All suspected deletions or duplications emerged through this software analysis were confirmed by MLPA.

### 4.3. Breakpoint Resolution of the LGRs

Primers for long-range PCRs for the respective LGRs were either designed in-house or adapted from publications (Appendix A). PCR reactions were done using Herculase II Fusion DNA Polymerase (Agilent Technologies, CA, USA). Amplification products were visualized on 1% agarose gels next to Hyper Ladder 1kb DNA sizing standard (Bioline, London, UK). PCR products containing the deletion or duplication breakpoints (that is, the junction fragments) were sequenced by conventional Sanger sequencing method on ABI3130 Genetic Analyzer (Thermo Fisher Scientific) using a series of nested walking primers to span the breakpoint. Sequencing primers are also listed in Appendix A. Exact designation of the LGR breakpoints was according to the current HGVS nomenclature recommendations [50]. Reference sequence LRG_292t1, which corresponds to NM_007294.3, was used for LGRs inside the *BRCA1* transcript and NG_005905.2 was used for LGRs extending beyond the boundaries of the *BRCA1* transcript towards upstream direction. Reference sequence LRG_293t1, which corresponds to NM_000059.3, was used for LGRs inside the *BRCA2* transcript. The detected LGR variants were uploaded to the LOVD v.3 Locus Specific Database [51] with accession numbers #296865-#296952.

Primer pairs for QMPSF were designed using the Primer3Plus software (primer sequences are available in Appendix A). One of the PCR primers was labeled with FAM fluorophore and each amplicon size was different. PCR reactions were performed by Qiagen Multiplex PCR Kit (Qiagen) with all primer pairs in one reaction tube. Fluorescent fragment analysis was done on ABI3130 Genetic Analyzer (Thermo Fisher Scientific) in microsatellite analysis method and the peaks were visualized by Peak Scanner Software 2.0 provided for the instrument. Height of each peak was compared to the respective peaks of a control sample after normalization. Amplicons for real time semi-quantitative PCR (qPCR) tests were designed individually for the interrogated regions (Appendix A). PCR reactions were set according to the instructions of the Brilliant HRM Ultra-Fast Loci Master Mix (Agilent Technologies) and run on AriaMX Real-time PCR System (Agilent Technologies) detecting FAM fluorescence in real time over 40 cycles. Two wild-type calibrator samples were run in each experiment. Amplicons designed for known one-copy regions and known two-copy regions were added as positive and negative controls, respectively. Results were analyzed by AriaMX Software provided for the instrument. Test regions with amplicons of ΔΔ Ct number equal 1 were regarded to be deleted in one copy. Individual PCR reactions were in triplicate and each test was confirmed in an independent reaction.

### 4.4. Haplotyping

Large genomic rearrangements, which affected the same exon(s) were subjected to haplotype analysis to reveal their possible common genetic origin. *BRCA1* haplotyping was done with short tandem repeat (STR) polymorphic microsatellite markers locating within the *BRCA1* gene and 50 kb surroundings: D17S1185, D17S1320, D17S1321, D17S855, D17S1322, D17S1323, D17S1327, D17S1326, D17S1325. PCR primer sequences were taken from https://www.ncbi.nlm.nih.gov/probe. PCR product fragments were detected on ABI3130 Genetic Analyzer in microsatellite analysis mode. GeneScan Liz500(-250) (Thermo Fisher Scientific) was used as sizing reference. Peaks were visualized using the Peak Scanner Software 2.0 and exact marker sizes were assigned for both alleles. Core haplotypes were determined in each case. Where it was applicable, additional information of other *BRCA1* variant genotypes, originating from the complete sequencing of the coding exons, was also incorporated in the whole genotype data of a sample and were taken advantage of for haplotyping.

### 4.5. RNA Isolation and RT-PCR

RNA was isolated by two different techniques: one applied the Tempus Spin RNA Isolation Kit (Thermo Fisher Scientific) from 9 mL peripheral blood taken into Tempus Blood RNA Tubes (Thermo Fisher Scientific) according to manufacturer’s instructions. For some samples, RNA was isolated using the NucleoSpin RNA Plus Kit (Macherey Nagel, Dueren, Germany) from peripheral blood mononuclear cells stored in RNALater (Sigma-Aldrich, Merck, Darmstadt, Germany), complying with the manufacturer’s recommendations. RNA quantity and quality were determined by NanoDrop ND-1000 Spectrophotometer (NanoDrop Technologies, Thermo Fisher Scientific), and 100–200 ng RNA was converted into cDNA by SuperScript IV Reverse Transcriptase (Thermo Fisher Scientific) using either random hexamers or oligo dT primers.

### 4.6. Allele Imbalance

Relative expressions of the variant and wild type *BRCA1* alleles were tested by measuring the Sanger sequencing electropherogram ratio of exonic heterozygote positions. Informative nucleotide positions were sequenced bi-directionally with flanking exonic primers on cDNA template. Analyzed sequencing data was visualized in Sequence Scanner program 2.0 (Thermo Fisher Scientific). Peak ratios for the heterozygote positions were compared to the peak ratios of gDNA sequence of the same position for the same sample. The relative ratios were calculated, and allelic imbalance was asserted if the difference was >50%. Information from several heterozygote positions were integrated, where it was applicable.

### 4.7. Statistical Analyses

Statistical analyses of phenotype parameters depending on the type of the data were performed either by two-tailed *t*-tests or Fisher’s exact tests. Numerical parameters, such as age at disease onset and Ki-67 status were compared by two-tailed *t*-tests. Categorical parameters, such as estrogen and progesterone receptor, Her2 IHC-status, presence of second primary tumor in the proband and positive family history were compared by Fisher’s exact tests. Difference was regarded at nominal significance *p* ≤ 0.05 at each comparison.

## Figures and Tables

**Figure 1 ijms-21-04650-f001:**
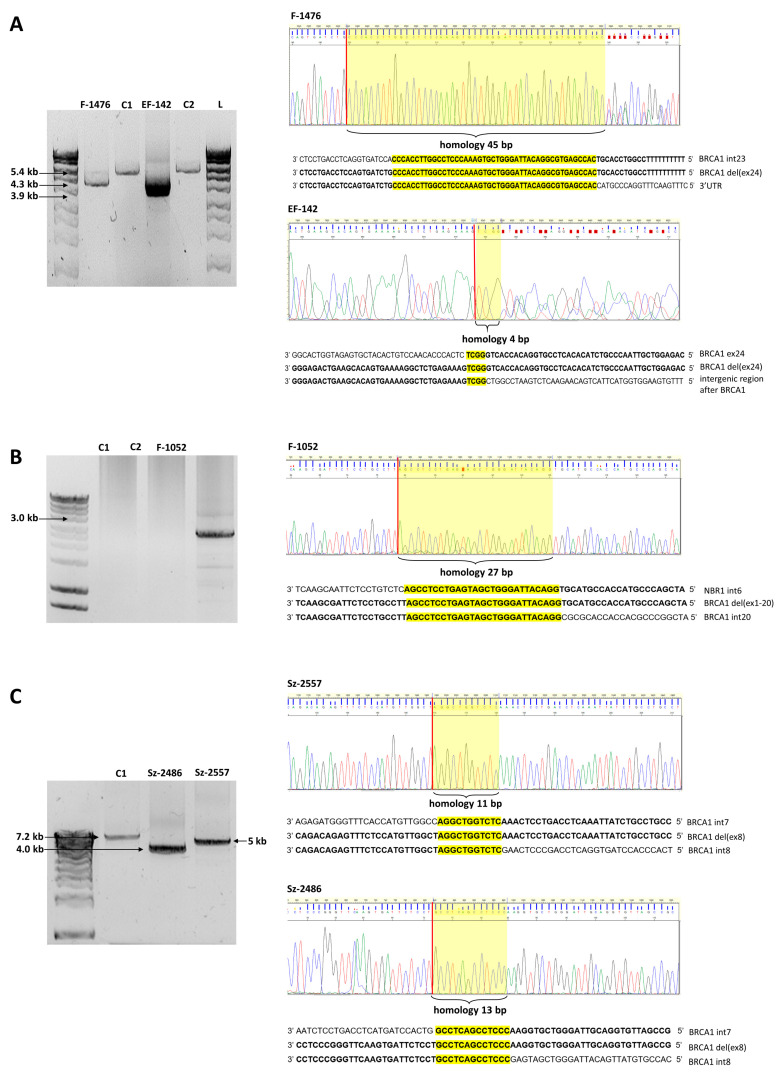
Representative examples of breakpoint determination in three rearrangement types. (**A**): *BRCA1* del(ex24) samples, (**B**): *BRCA1* del(ex1-20) sample, and (**C**): *BRCA1* del(ex8) samples. Upper panels: agarose gel visualization of the junction fragments. Lower panels: breakpoint spanning sequence electropherograms of the junction fragments. L: Hyper Ladder 1 kb DNA sizing standard. C1 and C2 are *BRCA1/2* negative control samples. Red bars represent the concerted breakpoints according to the 3’ rule. Yellow boxes highlight the perfect sequence homologies at the breakpoint junctions. Exact base sequences surrounding the breakpoints relative to the reference sequence are shown below the electropherograms. (**A**): F-1476 is a LRG_292t1:c.5468-364_*749del heterozygote sample with 1.1 kb deletion and EF-142 is a LRG_292t1:c.5506_*1383+36del heterozygote sample with 1.5 kb deletion. (**B**): F-1052 is a NG_005905.2:g.33502_166230del heterozygote sample with 133 kb deletion. The C1 and C1 samples did not yield PCR product of 136 kb. (**C**): Sz-2486 is a LRG_292t1:c.441+1521_547+392del heterozygote sample with 3.2 kb deletion and Sz-2557 is a LRG_292t1:c.442-1830_547+295del heterozygote sample with 2.2 kb deletion. The wild type alleles of the mutation carrier samples did not yield amplification products.

**Figure 2 ijms-21-04650-f002:**
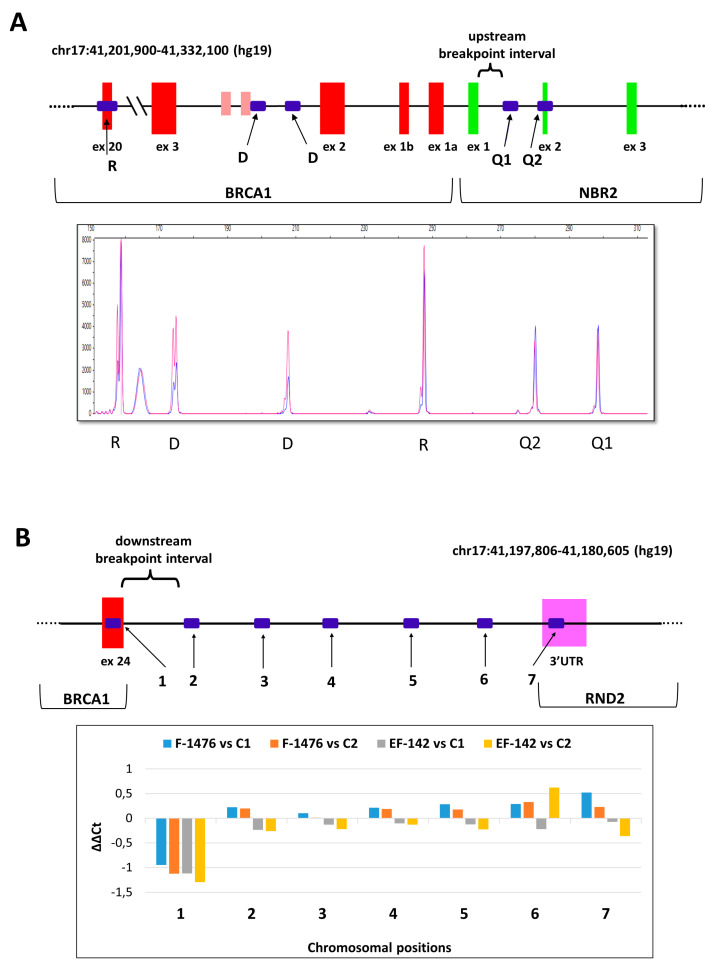
Narrowing down the breakpoint intervals with relative quantitation assays. Representative examples. (**A**): Quantitative multiplex PCR of short fluorescent fragments (QMPSF) assay of the *BRCA1* del(ex1-3) sample with variant NG_005905.2:g.(88971_92304)_(102259_111450)del (blue) vs. control sample (red). R sign denotes amplicon peaks for biallelic control positions, D sign denotes amplicon peaks for monoallelic control positions. Q1 and Q2 denote queried positions of the upstream breakpoint interval, which was formerly confined to this region by multiplex ligation-dependent probe amplification (MLPA). Both query positions were biallelic, thus deletion breakpoint should be downstream of these positions. The inferred breakpoint interval is highlighted with brace on the graphic of the tested chromosomal region beneath the electropherogram. Red bars are exons for *BRCA1*, green bars are exons for NBR2. (**B**): Real-time qPCR result for *BRCA1* del(ex24) samples with variants LRG_292t1:c.5468-364_*749del (F-1476) and LRG_292t1:c.5506_*1383+36del (EF-142). Four bars per amplicon represent the ΔΔCt values of each *BRCA1* del(ex24) samples relative to two different calibrators (C1 and C2). The standard errors of technical triplicates were below 5% for all measurements, so error bars are not shown. Position and numberings of the designed amplicon targets from *BRCA1* exon 24 towards downstream region of the gene is depicted on the graphic of the tested chromosomal interval beneath the chart. Red bars are exons for *BRCA1*, magenta bars are exons for *RND2*. Only the amplicon designed for *BRCA1* exon 24 showed deletion in both families, thus downstream deletion interval is restricted to the region between amplicons 1 and 2 (showed within brace).

**Figure 3 ijms-21-04650-f003:**
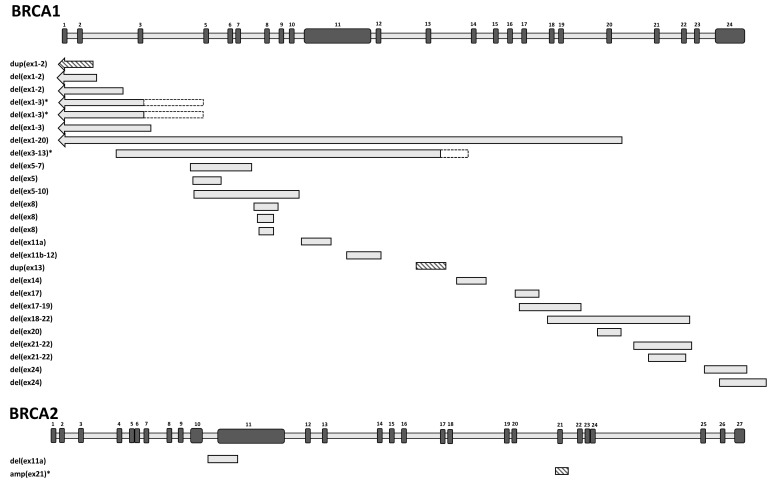
Position of the detected large genomic rearrangements (LGRs) along the *BRCA1* and *BRCA2* genes. Genes are drawn in sense orientation without promoter region. The length of exons (black bars) and introns (grey bars) are not exactly to scale. Exon numberings are given above them. Each different LGR variant is represented once. Deletions are depicted with grey boxes, duplications are depicted with striped boxes. Respective LGR running names indicating their exon affections are listed on the left. Arrows indicate that the rearrangements outreached the gene towards upstream direction. LGRs with uncertain breakpoints are labeled with an asterisk. Additionally, where breakpoint uncertainty is larger than 1 kb, dashed outlines of the bars indicate the possible breakpoint intervals.

**Figure 4 ijms-21-04650-f004:**
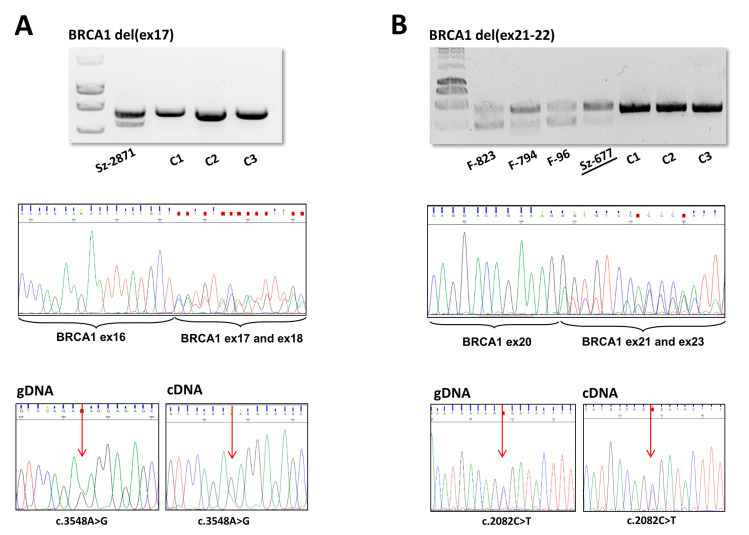
RT-PCR product, cDNA-level sequencing and allele expression ratio of (**A**): *BRCA1* del(ex17) and (**B**): *BRCA1* del(ex21-22) samples. PCR amplifications were done on cDNA templates with cDNA primers locating in exons flanking the rearrangements. Shorter fragments amplified from the deletion-carrier alleles were obtained in each case (upper boxes, agarose gel electrophoreses). Sz-2871 is a LRG_292t1:c.4986+726_5074+84del heterozygote variant carrier. F-823 is a LRG_292t1:c.5277+2114_5407-689del heterozygote variant carrier and F-794, F-96, Sz-677 samples are LRG_292t1:c.5278-492_5407-128delins236 heterozygote variant carriers. C1–C3: control wild type samples. Sequencing of the amplified regions resulted in canonical cassette exon junctions flanking the deleted regions in each family (Sanger sequence electropherograms in the middle boxes show the superposition of signals of the normal and deleted alleles from the position of the canonical exon borders). All *BRCA1* del(ex21-22) samples yielded the same sequencing result, Sz-677 (underlined) was represented as an example. Allelic ratio for elected exonic heterozygote positions (c.3548A>G and c.2082C>T, respectively, indicated by red arrows) were compared to that of the gDNA sequence according to the electropherogram AUCs (area under the curve) of the nucleotide superpositions (lower boxes). (**A**): *BRCA1* del(ex17) showed lower amount of transcript from the deleted allele. (**B**): *BRCA1* del(ex21-22) did not show cDNA allelic ratio difference relative to that the gDNA.

**Table 1 ijms-21-04650-t001:** List of the Hungarian large genomic rearrangements (LGR) variants.

Number of Probands	Gene	Running Name	CNV	HGVS Name	RNA	Protein	Reference	Upstream Breakpoint	Downstream Breakpoint	Homology	Inferred Rearrangement Mechanism
3	*BRCA1*	*BRCA1* dup(ex1-2)	dup	NG_005905.2:g.90060_97318dup	NA	NA	[7]	AluY	AluYk4	48 bp	NAHR
**1**	***BRCA1***	***BRCA1* del(ex1-2)**	**del**	**NG_005905.2:g.59988_96300del**	**NA**	**NA**	**NA**	**Psi*BRCA1* (AluSp)**	***BRCA1* intron 2 (AluSp)**	**15 bp**	**NAHR**
**1**	***BRCA1***	***BRCA1* del(ex1-2)**	**del**	**NG_005905.2:g.59885_96193del**	**NA**	**NA**	**NA**	**Psi*BRCA1***	***BRCA1* intron 2**	**62 bp**	**NAHR**
**1**	***BRCA1***	***BRCA1* del(ex1-3)**	**del**	**NG_005905.2:g.(24943_26402)_(102259_111450)del**	**NA**	**NA**	**NA**	**NA**	**NA**	**NA**	**NA**
**1**	***BRCA1***	***BRCA1* del(ex1-3)**	**del**	**NG_005905.2:g.(88971_92304)_(102259_111450)del**	**NA**	**NA**	**NA**	**NA**	**NA**	**NA**	**NA**
**1**	***BRCA1***	***BRCA1* del(ex1-3)**	**del**	**NG_005905.2:g.84958_106171del**	**NA**	**NA**	**NA**	**AluY**	**AluY**	**4 bp**	**MMEJ/SSA**
**1**	***BRCA1***	***BRCA1* del(ex1-20)**	**del**	**NG_005905.2:g.33502_166230del**	**NA**	**NA**	**NA**	**AluSx**	**AluSg**	**27 bp**	**MMEJ/SSA**
**1**	***BRCA1***	***BRCA1* del(ex3-13)**	**del**	**LRG_292t1:c.(81-2037_81-1)_(4357+1_4358-1)del**	**NA**	**NA**	**NA**	**NA**	**NA**	**NA**	**NA**
1	*BRCA1*	*BRCA1* del(ex5)	del	LRG_292t1:c.135-30_212+136del	NA	NA	[8]	Non-Alu	Non-Alu	9 bp	NHEJ
1	*BRCA1*	*BRCA1* del(ex5-7)	del	LRG_292t1:c.135-1004_441+1608del	NA	NA	[8]	AluSz6	AluSc5	15 bp	MMEJ/SSA
1	*BRCA1*	*BRCA1* del(ex5-10)	del	LRG_292t1:c.135-4505_670+361delins35	LRG_292t1:r.135_670del	p.(Lys45AsnfsTer3)	[3]	AluSx/AluY	AluY/AluJb	10 bp/29 bp	FoSTeS
**2**	***BRCA1***	***BRCA1* del(ex8)**	**del**	**LRG_292t1:c.441+1521_547+392del**	**LRG_292t1:r.442_547del**	**p.(Gln148AspfsTer50)**	**NA**	**AluSc5**	**AluSp**	**13 bp**	**MMEJ/SSA**
2	*BRCA1*	*BRCA1* del(ex8)	del	LRG_292t1:c.442-1102_547+252del	NA	NA	[24]	AluSx	AluSp	26 bp	MMEJ/SSA
**1**	***BRCA1***	***BRCA1* del(ex8)**	**del**	**LRG_292t1:c.442-1830_547+295del**	**NA**	**NA**	**NA**	**AluSz**	**AluSp**	**11 bp**	**MMEJ/SSA**
**1**	***BRCA1***	***BRCA1* del(ex11a)**	**del**	**LRG_292t1:c.671-216_901del**	**NA**	**NA**	**NA**	**non-Alu**	**non-Alu**	**4 bp**	**NHEJ**
1	*BRCA1*	*BRCA1* del(ex11b-12)	del	LRG_292t1:c.1644_4185+3618del	NA	NA	[25]	non-Alu	L1	3 bp	NHEJ
8	*BRCA1*	*BRCA1* dup(ex13)	dup	LRG_292t1:c.4186-1787_4357+4122dup	NA	NA	[26]	AluSx	AluSx1	23 bp	unknown
**1**	***BRCA1***	***BRCA1* del(ex14)**	**del**	**LRG_292t1:c.4357+1661_4485-338del**	**NA**	**NA**	**NA**	**AluJo**	**AluSx1**	**10 bp**	**MMEJ/SSA**
4	*BRCA1*	*BRCA1* del(ex17)	del	LRG_292t1:c.4986+726_5074+84del	LRG_292t1:r.4987_5074del	p.(Val1665SerfsTer7)	[27,28]	AluSp	AluSc	10 bp	MMEJ/SSA
1	*BRCA1*	*BRCA1* del(ex17-19)	del	LRG_292t1:c.4987-365_5194-484del	NA	NA	[29]	AluY	AluY	43 bp	MMEJ/SSA
1	*BRCA1*	*BRCA1* del(ex20)	del	LRG_292t1:c.5213_5278-2753delinsA †	NA	NA	[30]	non-Alu	AluSp	no	NHEJ
2	*BRCA1*	*BRCA1* del(ex18-22)	del	LRG_292t1:c.5075-1135_5406+346del	LRG_292t1:r.5075_5406del	p.(Asp1692GlyfsTer26)	[3]	AluY	AluSz	7 bp	MMEJ/SSA
9	*BRCA1*	*BRCA1* del(ex21-22)	del	LRG_292t1:c.5278-492_5407-128delins236	LRG_292t1:r.5278_5406del	p.(Ile1760_Thr1802)	[25]	non-Alu/AluSx	AluSx/AluJb	26 bp	FoSTeS
**1**	***BRCA1***	***BRCA1* del(ex21-22)**	**del**	**LRG_292t1:c.5277+2114_5407-689del**	**LRG_292t1:r.5278_5406del**	**p.(Ile1760_Thr1802)**	**NA**	**AluSq2**	**AluSc**	**8 bp**	**MMEJ/SSA**
**1**	***BRCA1***	***BRCA1* del(ex24)**	**del**	**LRG_292t1:c.5506_*1383+36del**	**NA**	**NA**	**NA**	**non-Alu**	**non-Alu**	**4 bp**	**NHEJ**
**1**	***BRCA1***	***BRCA1* del(ex24)**	**del**	**LRG_292t1:c.5468-364_*749del**	**NA**	**NA**	**NA**	**AluSx**	**AluSx1**	**45 bp**	**MMEJ/SSA**
**1**	***BRCA2***	***BRCA2* del(ex11a)**	**del**	**LRG_293t1:c.1910-92_3888del**	**NA**	**NA**	**NA**	**non-Alu**	**non-Alu**	**no**	**NHEJ**
**1**	***BRCA2***	***BRCA2* amp(ex21)**	**amp**	**LRG_293t1:c.(8633-70_8633-1)_ (8754+78_8754+122)amp**	**NA**	**NA**	**NA**	**NA**	**NA**	**NA**	**unknown**

Designations of the LGRs are according to the current HGVS (Human Genome Variation Society) nomenclature. † This variant was described by Belogianni et al., 2004 [30], with the exception that we found an additional adenine residue inserted at the junction position. A shorter running name is given for rearrangement types for easier reference in the text. Novel LGRs, reported first in this article are highlighted with bold. Reference is given for formerly described LGRs. RNA nomenclature and inferred protein effects are determined, where cDNA-level sequencing was performed. NAHR: non-allelic homologues recombination, NHEJ: non-homologues end-joining, MMEJ/SSA: microhomology-mediated end-joining/single-strand annealing, FoSTeS: fork stalling and template switching.

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
