# Peer review of "Complex Characterization of Germline Large Genomic Rearrangements of the BRCA1 and BRCA2 Genes in High-Risk Breast Cancer Patients—Novel Variants from a Large National Center"

_ijms, 2020, doi:10.3390/ijms21134650_

Round 1
Reviewer 1 Report
The authors describe LGRs identified in Hungary with the great majority occurring in BRCA1 and accounting for 10% of BRCA1 pathogenic variants but <1% of BRCA2. The paper is well written and the analysis has been well executed. The absence of a genotype phenotype is expected but still a useful addition. It would help in interpreting the results to know if there are founder effects for point mutations in Hungary
Specific points
- ’ but fall short of the extreme ratios 194 found in the Netherlands (36% of all BRCA1 mutants) [14] and in Italy (19% of all BRCA1 mutants) 195 [15], where frequently occuring founder mutations constitute the majority of LGR cases.’ –Occurring spelt incorrectly. Also UK has as high as 20% in non-Jewish outbred UK population.
- ‘BRCA2 was nearly exempt from 197 LGRs. This is in concert with the findings worldwide.’ –Please give frequency range for BRCA2 form worldwide literature.
- Please comment on whether NGS dosage is a reliable test to identify the LGRs identified? Where they all found on MLPA?
- Please provide more information on point mutations found in your lab and whether founder effects for these may explain the relatively lower proportions of LGRs
- A minor point but the page numbering is faulty
Reviewer 2 Report
Bozsik et al. performed a characterisation of large genomic rearrangements (LGRs) in the BRCA1 and BRCA2 gene loci in breast and ovarian cancer patients. The families for this study were enrolled for BRCA1/2 genotyping at the National Institute of Oncology, Hungary, and once deemed eligible, the detection of whole exon deletions and duplications was performed by Multiplex Ligation-dependent Probe Amplification (MLPA). The authors found 28 different types of LGRs in 51 probands analysed. Out of these 28 LGRs, 16 had not been reported before. The LGRs identified consisted of 3 exon copy gains, 2 complex rearrangements and 23 exon losses. The scope of the article was:
- to determine the upstream and downstream breakpoint for each rearrangement
- to perform a transcript-level examination of the variant allele
- to investigate the molecular mechanisms that caused the presence of LGRs
- to perform a haplotype analysis to reveal a possible common genetic origin of LGRs
- to investigate the pathogenicity of the LGRs in BRCA1 and BRCA2 genes
Overall, this article presents novel large genomic rearrangements in the BRCA1 and BRCA2 genes to the scientific community in a unique cohort of Hungarian patients.
Major comments:
1. It is highly recommended that the quality of the figures in this article be improved. Some appear to be screenshot images, which unfortunately are not up to standard for publication. Each figure in the paper should include clear labels of the axes, colour-coded legends, error bars and number of replicates. Some of the numbers and letters in the figures are not readable, even after zooming in.
2. The figures chosen for this article only represent examples of the LGRs found in this study. The authors are recommended to add supplementary figures that portray the experimental results for each of the LRGs depicted in Table I. Alternatively, or in addition, they could feature more examples in the main figures.
Minor comments:
1. Misspellings have been found throughout the article.
2. References need to be added on lines [35-37].
3. In Table I, can the authors indicate the number of individuals carrying a specific LGR instead of the number of families?
4. The “Exon” column in Table I is not necessary because the exons involved in an LGR are depicted in the “Running name” column.
5. In Table I, it would be helpful to include a column indicating if the families in the cohort are either ovarian or breast cancer related families.
6. In Table I, abbreviation explanations are missing in the figure legend.
7. In Figure 1, can the authors describe in more detail the controls that they have used for the agarose gel visualisation of the junction fragments?
8. A clearer link between the homology sequence and the junction fragment of an LGR is necessary in the main text. In Figure 1 (bottom), it would be even more informative to highlight the deleted bases from exon 24 as well as the homology sequence to distinguish between the two of them.
9. In Figure 1B, the authors do not indicate the number of replicates, the error bars are not included and the labelling of Y/X axes are missing. The figure would also benefit by the addition of a legend description for each included element.
10. In Figure 2, can the authors move the genomic representations of the breakpoint intervals in the upper part of each panel to help to reader?
11. In Figure 2A, the authors should clarify how they choose to investigate exactly the Q1 and Q2 regions. If this figure represents the deletion of exon 1-3 of BRCA1, why didn’t the authors consider investigating a breakpoint between exon1a of BRCA1 and exon1 of the NBR2 gene?
12. Why are the numbers on the X- and Y-axes of the upper panel of Figure 2A inverted?
13. Why is exon20 of the BRCA1 gene underlined in Figure 2A?
14. It would be more clear if the genes to which the exons belong to are written in each panel of Figure 2.
15. In line 85 of the main article text, the authors mentioned that “In all but four cases we managed to determine the exact breakpoints of LRGs”. However, in Figure 3, there are only 3 LGRs with dashed lines, which represent the ambiguous breakpoints as specified in the figure description. Is 1 LGR missing a dashed outline in Figure 3?
16. In Figure 4, can the authors provide the name of the deletion at the top of both panel A and B?
17. From Table I, the BRCA1 del(ex17) was detected in 4 families, hence it was detected in at least 4 individuals. Why have the authors decided to show only the Sz-2871 individual?
18. In Table I, there are 2 different BRCA1 del(ex21-22) LRGs. Hence, the authors should specify which one they are considering in particular for panel B of Figure 4.
19. In Figure 4, numbers and letters are not easily read.
20. The nucleotides should always be visible in the bottom panels of Figure 4 and the red arrows should be labelled in the figure as well as in the figure description.
21. In the supplementary Figure 1 description, the authors should clarify how many people out of the 24 samples analysed they found the 5 deleted exons.
22. Both panels of the supplementary Figure 1 require labels on the axes and a colour-coded legend.
23. In Supplementary Table III, can the authors explain why the total number of breast cancer patients in the “non-LGR column” at the “tumour-hormone status” rows changes?
24. In lines 149-152, the authors claim that “where deletion stretched further upstream the BRCA1 gene affecting also the BRCA1 and NBR2 common promoter or even the NBR2 or NBR1 genes themselves did not show more severe disease pathology than LGRs restricted to the BRCA1 open reading frame”. Would it be possible to see the experimental results that support this statement?
Round 2
Reviewer 2 Report
The authors addressed all my comments.